# UAV-Based High-Rise Buildings Earthwork Monitoring—A Case Study

**Hyung Cheol Park, Titi Sari Nurul Rachmawati and Sunkuk Kim \***

Department of Architectural Engineering, Kyung Hee University, Yongin-si 17104, Korea
\* Correspondence: kimskuk@khu.ac.kr; Tel.: +82-31-201-2922

**Abstract:** Unmanned aerial vehicle (UAV) is one of the most prominent technologies in the construction industry for data collection purposes. Compared with traditional methods, UAVs collect data faster and more efficiently at a lower cost. One of the construction works that can be monitored using UAV is earthwork. Earthwork monitoring is essential to complete the earthwork on time, according to plan, and within budget. This paper presents an application study on the UAV-based earthwork monitoring of a high-rise building project in the Republic of Korea. Earthwork of building projects have distinct characteristics. The area is excavated downwards to tens of meters deep, thus contractors deal with several types of soil. The building project is usually built in a densely built area. Therefore, contractors must monitor the slope as it poses landslide risk to surrounding areas. UAV can calculate the excavated volume, monitor the progress and the site, and document earthwork periodically and strategically. Based on case study, this study compared estimated volume based on GPS and actual excavated volume based on UAV survey and found 0.71% difference, indicating the reliability of surveying using UAV. However, the volume per soil type was quite different between both methods, resulting in 15.8% (USD 183,057) cost difference. This study shows that UAV technology is effective in monitoring the actual excavated volume, thus supporting fair business practices and transparency between stakeholders.

**Keywords:** unmanned aerial vehicle (UAV); earthwork management; quantity survey; site monitoring

## 1. Introduction

Earthwork is a fundamental process in the early stage of construction projects. Earthwork consists of any construction work carried out on the Earth's surface, including clearing, grading, excavating, compacting, cutting, filling, and finishing [1]. During earthwork, the topography of site changes rapidly. Contractors must monitor the continuous changes over the construction period. Topographical change data are highly important as they serve as the main data for earthwork design review, progress monitoring, estimating earthwork volume, and documentation [2]. Moreover, contractors also must pay attention to the earthwork safety and quality standards.

An accurate and quick method to survey the earthwork topography is required so that analysis, prediction, monitoring, and documentation can be performed. Two types of surveying methods can be used to acquire the topographic information at earthwork sites [3]. The first type is a point-based method which consists of total station (TS) and global positioning system (GPS) technology. The second type is an area-based method which consists of unmanned aerial vehicle (UAV) photogrammetry and laser scanning.

Several studies compared these topographic surveying methods. Beretta et al. [4] compared UAV with GPS survey and laser scanning. The UAV results were less erratic than laser scanning and offer lower uncertainty than GPS survey. According to Akgul et al. [5], UAV is more efficient than GPS for surveying larger areas while also producing accuracy that complies with surveyor requirements. Additionally, UAV surveying produces both a digital surface model (DSM) and georeferenced ortho-mosaic, which is an important added

value for site documentation [6]. Overall, UAV has higher efficiency than conventional surveying methods in terms of cost, data acquisition, and processing while also producing equivalent accuracy [4]. UAV is one of the selected technologies included in the vision of "Smart Construction 2025" of South Korea [2]. Accordingly, Korean National Geographic Information Institute (NGII) published "Guidelines for the Public Survey using UAV" in 2018, which contains standard procedures for conducting UAV photogrammetry to public surveys at construction sites [7]. It is expected that UAV usage will increase significantly in the future.

Although there are several studies studying the use of UAVs for earthwork, the scope of the project is limited to road construction [2,8] and stockpile volume calculations [9,10]. There are no studies investigating the use of UAVs for earthwork in high-rise building projects. Building projects have different characteristics from other projects. Building projects are usually built in densely built areas so that the earthwork process must pay attention so as not to damage or cause landslides to surrounding buildings. In addition, the area is excavated downwards to tens of meters deep, posing a risk of landslides. The slope must be monitored to avoid landslides and accident to workers [11]. Moreover, with a depth of tens of meters, contractors deal with several types of soil with different characteristics. Earthwork also takes place over a long period of time, so the volume of earthwork needs to be monitored regularly. Volume monitoring serves not only as data for payment progress but also to avoid the risk of cost overruns.

UAV as a data collection technology of terrain model has a role in increasing the efficiency of earthwork monitoring. Therefore, this study presents UAV-based earthwork monitoring and digitization of a high-rise building project. This study incorporates an actual apartment complex project in the Republic of Korea. UAV-based earthwork monitoring mainly consists of data collection, data processing, and data analysis and visualization. The framework is expected to digitize and assist earthwork volume calculation, progress monitoring, site monitoring, and documentation.

## 2. Theoretical Framework

### 2.1. Overview of Earthwork and UAV Roles in Earthwork

Earthwork is one of the most crucial work packages in building projects. Earthwork consists of soil surveying, measuring, and planning before the actual earthmoving. In the construction stage, the subcontractors bring heavy equipment such as excavators, dozers, rollers, graders, and dump trucks to the sites to perform earthwork clearing, excavating, and compacting and ultimately finish the desired design specifications [12]. In general, in order for excavation to be carried out, it is necessary to build a retaining wall with ground anchors to retain the soil. Installation of retaining walls cannot be performed haphazardly, especially if the project is carried out in a densely built area [13].

Despite the great scale of earthmoving, earthwork requires work precision, as any misalignment of the earthwork can result in time-consuming correction [14]. Therefore, earthwork progress monitoring, quality control, documentation, and site monitoring is highly important. The verification and design checks of earthwork heavily rely on the production of DSM [14]. Few surveying technologies to produce DSM have been established, namely TS, laser scanning, GPS, and UAV. Although TS and GPS can generate accurate global coordinates of measured points, generating a high-density 3D model of large areas with them is challenging. This is because TS and GPS methods require workers to actively move in the field, which is labor-intensive and time-consuming [5]. In addition, changes in contours and depth of survey area adds difficulty and risk level for TS and GPS surveyors [15]. Laser scanning can create high-density 3D point clouds but it takes plenty of time to process the data. Laser scanning is also prone to occlusions due to slope crests or other objects which can cause errors to the produced terrain surface [4].

In comparison with other previous technologies, UAV provides more comprehensive functions related to 3D topographic models. UAVs serve many purposes following those needed in earthwork, such as earthwork surveying [16], on-site management [17], progress

monitoring [18], and safety inspections [19]. Harwin and Lucieer [20] compared UAV-based volume calculations with conventional methods and showed that UAVs offer equivalent accuracy. Another study by Hugenholtz et al. [21] concluded that UAV-based volume calculations are more cost-effective and efficient than conventional methods. UAVs can reach areas that are more expansive and inaccessible to human workers. Furthermore, the data from UAVs are easily stored, documented, and can be used for Building Information Modeling input, which can improve communication and evaluation of the construction progress between stakeholders [22].

In general, the camera mounted in the UAV is used for taking images of construction sites [23]. These images are processed using the photogrammetry technique which provides a DSM of the area as one of the results. A comparison of the current 3D terrain model with the planned 3D terrain model is performed periodically to monitor the progress and review the design. Moreover, the excavation volume and height differences can be calculated automatically and quickly [5]. By monitoring the DSM, the contractor can determine whether the earthwork is progressing according to the design plan, and make adjustments accordingly [24]. These data also act as the main reference for calculating progress payments from the main contractor to the earthwork subcontractor.

### 2.2. Earthwork Life Cycle

As shown in Figure 1, the earthwork process was divided into design, procurement, and construction phases. A specific bill of quantities (BQ) was produced in each phase. A BQ is a document containing work items and their materials, equipment, and labor [25]. The material, equipment, and labor are measured by the corresponding number, area, volume, weight, or time. Lastly, each work item is priced, and the cost of all work items is summed as the total project cost. BQs are predominantly used as tender documents that assist contractors in pricing construction projects and as contract documents that serve as agreements between the involved parties [26].

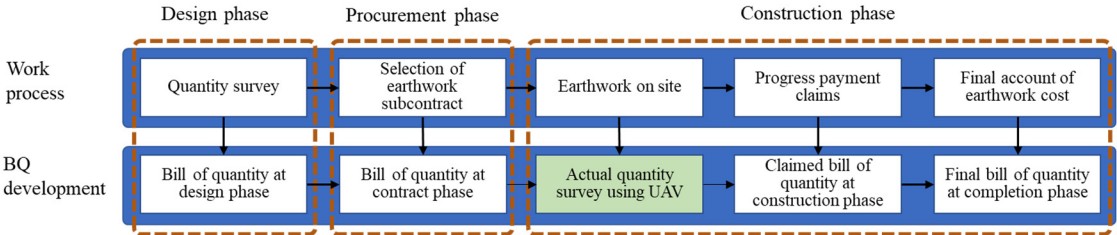

**Figure 1.** Earthwork process.

In the design phase, initial DSM was created based on the digital elevation map from the NGII which is officially managing digital elevation models for public use [27]. The volume calculation is roughly calculated by comparing the initial DSM with the planned surface. As a result, the earthwork volume that needs to be excavated is written in a BQ at the design phase.

The main contractor bids to select an earthwork subcontractor between the design and procurement phases. It is common for the main contractor to open a bid on specialized work such as earthwork in order to manage their schedules, quality, and budget to minimize the project's risk. Another reason for using a competitive process is to ensure that the subcontractors are specialized entities that are experts in their work fields and can meet the main contractor's requirements [28].

The subcontractor candidates conduct their soil investigations and produce a BQ as one of the tender documents. Subsequently, the chosen subcontractor produces the BQ at the contracting phase. In this phase, the earthwork subcontractor conducts a GPS survey and boring test. The GPS survey is performed to obtain the current DSM while boring test is conducted to obtain the soil stratification per boring test location [29]. By interpolating boring test data, model of surface layers per soil type is obtained [30]. By combining initial

DSM, surface layers per soil type, and the planned surface, estimated volume per soil type is obtained. This result is expected to be more precise than those in the BQ from the design phase.

The subcontractor executes earthwork packages in the construction phase, including supplying the workers, equipment, tools, designs, and other supplies following the contract documents [31]. In this stage, an actual quantity survey of earthwork is conducted using surveying methods such as UAV technology. Periodically, the subcontractor releases a progress report called "claimed BQ" to the contractor to claim progress payments. Additionally, when a modification to the earthwork's design, quality, or quantity affects the contracted sum, the subcontractor states this change in the progress report. The contractor then checks the "claimed BQ", especially the volumetric calculations, to process the progress payment. In this phase, there are sometimes disputes over volume calculations being overclaimed by the subcontractor or underclaimed by the contractor. Thus, a reliable volume calculation of the earthwork must be developed to ensure fair business dealings between stakeholders.

Ultimately, the subcontractor releases the final BQ after completing construction, resulting in all parties knowing the total earthwork cost. The final BQ contains financial components such as a statement of the final account, a final account summary, and adjustments that have been made.

## 3. Study Area and Methodology

### 3.1. Study Area

In this research, we present a case study of a construction site. Figure 2 shows that the construction project used for this case study was an apartment building complex in Seoul, Republic of Korea. The total land, building area, and total floor area are 33,897, 7181 and 121,716 m$^2$, respectively. The building-to-land ratio is 21%, and the floor area ratio is 227%. The project consists of ten buildings, and each building consists of 5 floors below ground and 18 floors above ground. The buildings comprise a residential area with 771 households in total. A high-density residential area surrounds the construction site. In addition, there is hilly terrain on one side of the site.

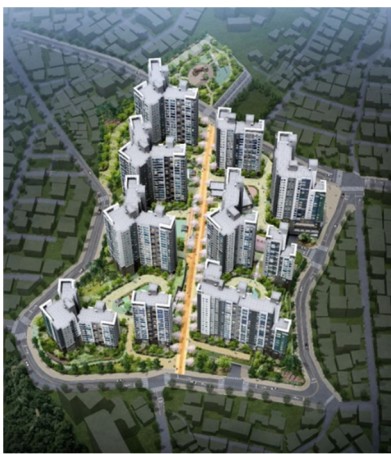

**Figure 2.** Case study of an apartment building complex project.

The earthwork ran from 16 February to 23 December 2021, as shown in Figure 3. Here, $t_0$ represents the start of construction. There were three activities conducted between $t_0$ and $t_1$: scheduling of the work and engineers, quantitative surveying of the earthwork measured by GPS, and open bidding to select the earthwork subcontractor. Subsequently, the subcontractor was chosen, and earthwork started at $t_1$. The quantity measured by GPS refers to the quantitative estimation using the GPS method and boring test results. Based on boring test, the soil at the construction site was composed of three types: (1) soil and weathered soil, (2) weathered rock, and (3) soft and hard rock. The contract proceeded with

the quantity measured by GPS but was finalized based on the quantity measured using UAV after completion.

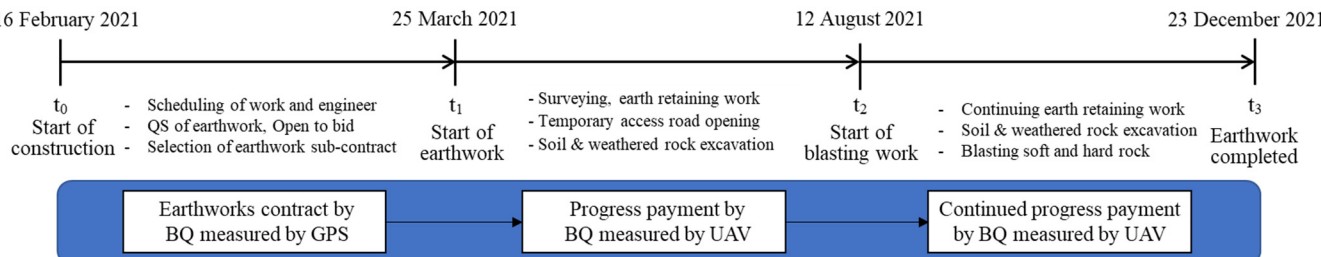

**Figure 3.** Earthwork timeline of the apartment building project.

Between $t_1$ and $t_2$, a temporary access road was opened, and earth surveying was performed. The soil and weathered rock were excavated first to uncover the underground soft and hard rock. Then, the soft and hard rock were blasted from the $t_2$ phase. Between $t_2$ and $t_3$, the subcontractor continued the earth retaining work and excavated the rest of the soil and weathered rock. From $t_2$ to $t_3$, the quantity of worked earthwork was periodically measured using UAVs. Since the earthwork quantity measured by UAVs was the actual excavation, it was used for the progress payment.

### 3.2. Methodology

This study proposes a framework for the earthwork monitoring and digitalization of building construction sites based on UAVs, as shown in Figure 4. This framework consists of three stages: (1) data collection, (2) data processing, and (3) data analysis and visualization. In the first stage, the UAV is used to acquire aerial pictures of the site, which is performed periodically. The images are processed using the Pix4Dmapper software in the second stage, where an ortho-mosaic and 3D point cloud are generated. Finally, in the third stage, the ortho-mosaic and 3D point cloud are used to generate cross-section views and calculate the cut-and-fill volume. The cross-section views and cut-and-fill volume are visualized in the UAV platform. The data and visualization stored in the UAV platform can be used for progress monitoring and documentation. Lastly, 2D/3D visualization can be used for site monitoring.

#### 3.2.1. Data Collection

In this study, UAV is employed for earthwork data collection every 3–5 days. A commercial quadcopter UAV DJI Phantom 4 Pro V2.0 with a 20-megapixel camera was used [32]. The weight and size of the UAV were 1375 g and $289.5 \times 289.5 \times 196$ mm$^3$, respectively. The UAV, internet, and system screen were prepared. The UAV condition, including battery capacity, GPS reception, IMU, and calibration status, and connection with the controls, was checked. In the Republic of Korea, UAV flight must obey the Act on Promotion of Utilization of Drones and Creation of Infrastructure [33]. Therefore, we consulted with the military regarding the no-fly zone to gain permission for the UAV flight. We also ensured no risk of collision with high-rise buildings, high mountains, or nearby forests. In addition, we also checked the weather where flight is aborted during windy and rainy weather. Then, we set the flight settings. The UAV flight plan is shown in Figure 5. The overlapped images were set to more than 80%. By collecting more overlapped images, we gained higher accuracy. We also set the UAV's speed to 8–10 m/s after considering the flight route and the UAV's maximum flight time. The UAV flew for approximately 30 min. Finally, after the flight was completed, the images were taken from UAV's SD card to be processed in the data processing stage.

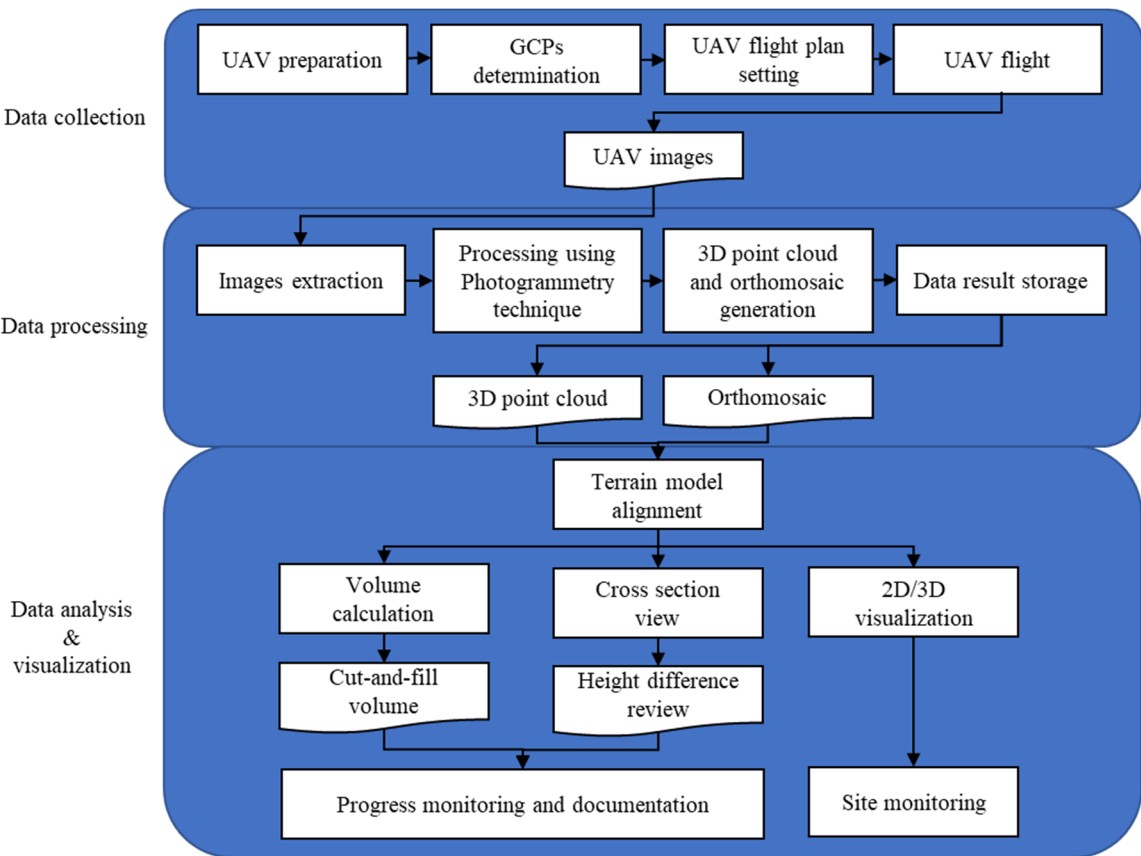

**Figure 4.** UAV-based earthwork management framework.

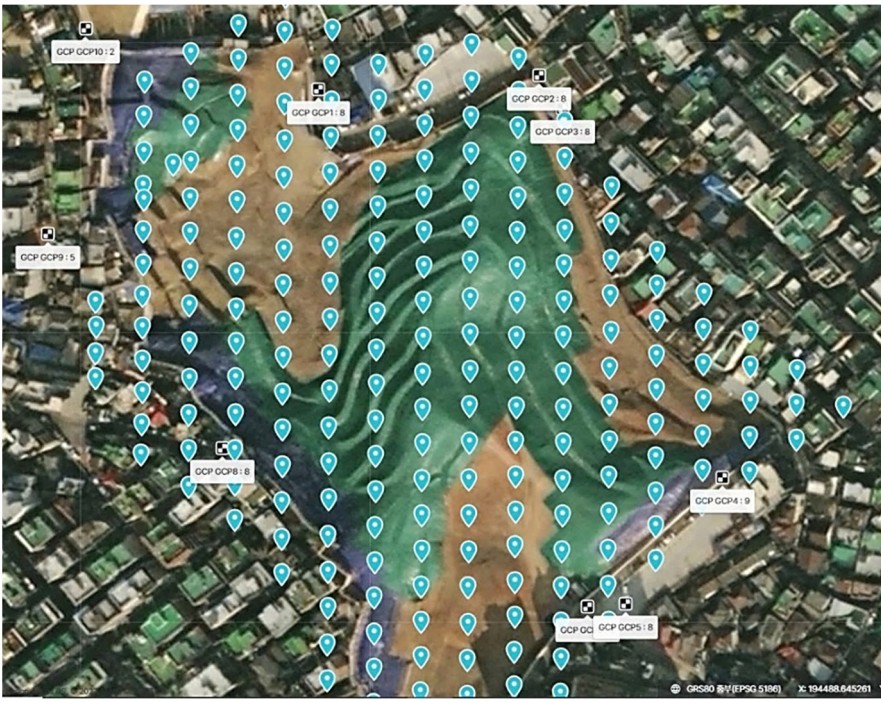

**Figure 5.** UAV's flight plan.

Ten ground control points (GCPs) were installed for georeferencing purpose of the UAV images. The installation of the GCPs was carried out according to a guideline [34]. GCPs were located outside the earthwork site and appropriately marked to be easily

detected in the UAV images as shown in Figure 6. Real-time kinematic global positioning system (RTK-GPS) was used to measure the latitude and longitude of GCPs. Meanwhile, the altitude was directly measured using RTS. The coordinate as shown in Table 1 was based on the GRS80 central origin (EPSG:5186). For the GCPs, it was confirmed that the accuracy of the coordinates is less than 3 cm in x and y-direction, and 5 cm in z-direction.

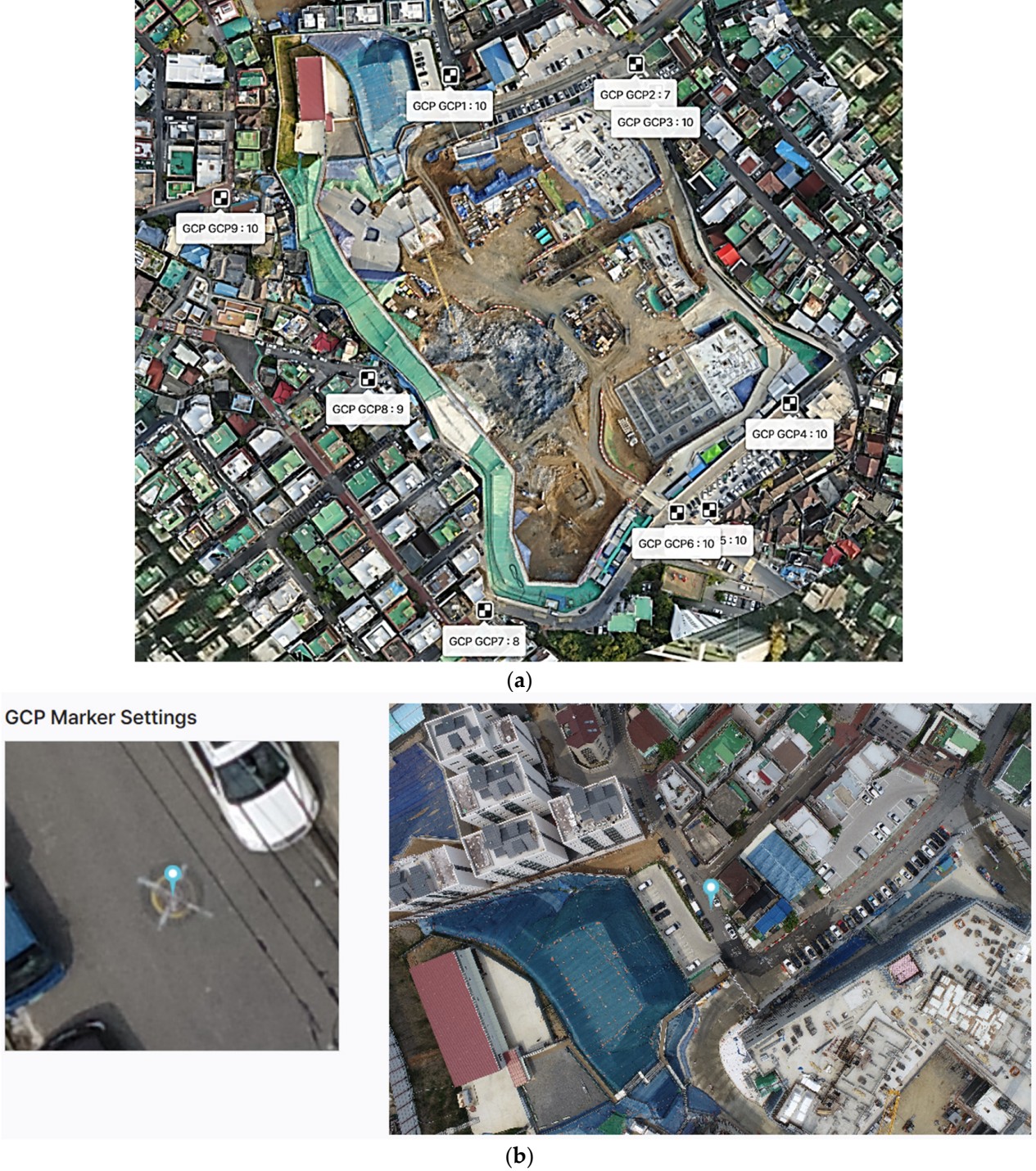

**Figure 6.** Ground control points of the project: (**a**) GCP locations; (**b**) location of GCP Number 1.

**Table 1.** Coordinates of the ground control points.

| GCP | x (m) | y (m) | z (m) |
| --- | --- | --- | --- |
| GCP1 | 194,565.142 | 544,356.834 | 41.643 |
| GCP2 | 194,657.481 | 544,362.629 | 38.151 |
| GCP3 | 194,667.442 | 544,348.826 | 38.775 |
| GCP4 | 194,734.263 | 544,194.89 | 54.503 |
| GCP5 | 194,693.649 | 544,142.25 | 59.505 |
| GCP6 | 194,677.789 | 544,140.788 | 60.644 |
| GCP7 | 194,582.005 | 544,092.945 | 75.396 |
| GCP8 | 194,524.529 | 544,207.286 | 73.361 |
| GCP9 | 194,451.559 | 544,296.509 | 64.822 |
| GCP10 | 194,461.096 | 544,440.59 | 60.216 |

### 3.2.2. Data Processing

The data processing in this study followed a photogrammetry workflow. First, the UAV images were imported from data storage into the photogrammetry software. In this study, the Pix4Dmapper software was used [35]. Images with the same georeferenced points were aligned, resulting in determination of the images' relative locations. The aligned images, which contained the GPS coordinates from the UAV, were then referenced to the GCPs. Then, a dense point cloud was generated using the referenced images and information about the optical system of the UAV (sensor size, focal length, angular resolution, and lens distortion) [36]. The 3D point cloud is shown in Figure 7. The individual points from the dense point cloud were then connected to form a mesh. From this mesh, an ortho-mosaic was created. Finally, the processed data were sent back to data storage. The point cloud was saved as an XYZ file, i.e., a list of each point with its X, Y, and Z coordinates and other properties of the point. The ortho-mosaic was stored as an uncompressed TIFF file to preserve all details. Although, the accuracy of UAV-based point cloud map was not analyzed, a standard UAV flight procedure, as described earlier (e.g., number of GCPs, location of GCPs, flight settings, etc.) was strictly followed [14].

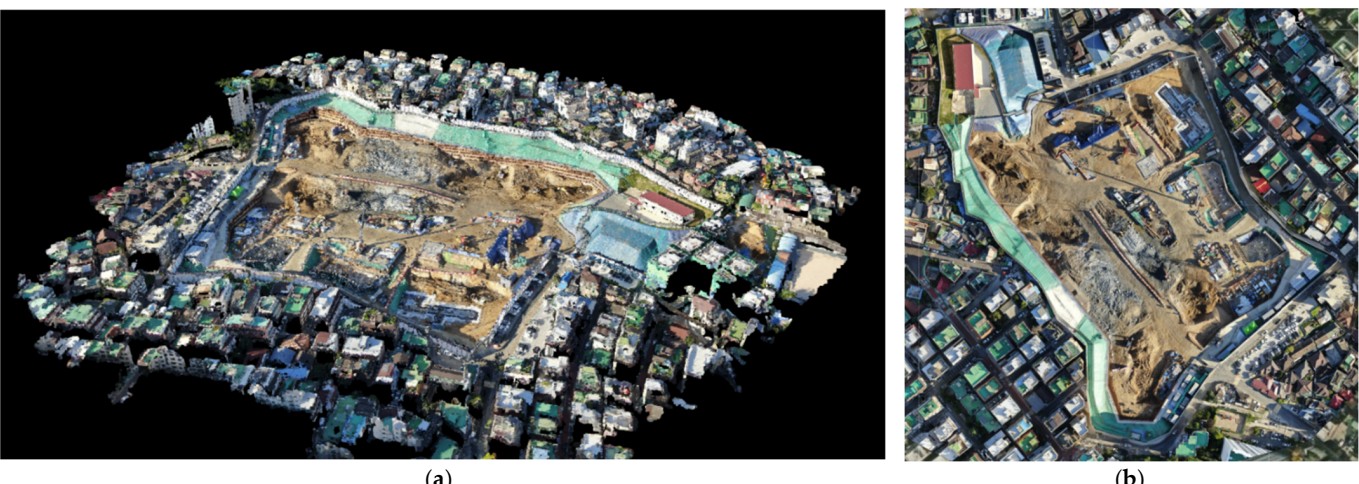

(**a**)                      (**b**)

**Figure 7.** Data processing results: (**a**) 3D point cloud; (**b**) ortho-mosaic.

### 3.2.3. Data Analysis and Visualization

The 3D models resulting from the data processing stage were visualized using the UAV platform which can be accessed remotely anytime [37]. There are four primary analysis and visualization types that can be performed on this platform: (1) automatic volume calculation with cut-and-fill volume data, (2) height difference review by comparing two terrain models from different time stamps, (3) site monitoring through 2D/3D visualization, and (4) documentation of the project from start to completion.

## 4. Results and Discussion

### 4.1. Volume Assessment

In this study, the earthwork quantity was measured using GPS and UAV technology. The GPS survey was performed in the planning stage to estimate the earthwork quantity. Meanwhile, a UAV-based survey was conducted to obtain the actual excavation. The measured quantity is shown in Table 2. The GPS's computed volume was 354,399 m$^3$, while the calculated volume based on UAV data was 351,883 m$^3$. There was a 2516 m$^3$ (0.71%) difference between the two surveying techniques, which is very small. The difference shows the reliability of surveying using UAVs. This finding is similar to Raeva et al. [10] and Mantey and Aduah [38] where UAV photogrammetry is an accurate method for calculating earthwork volume and more efficient than GPS.

**Table 2.** Quantity comparison measured by GPS and UAVs.

| Soil Layer | Quantity (m$^3$) | | Quantity Difference (m$^3$) |
|---|---|---|---|
| | **GPS** | **UAV** | |
| Soil and weathered soil | 222,541 | 235,103 | −12,562 |
| Weathered rock | 52,212 | 55,226 | −3014 |
| Soft and hard rock | 79,646 | 61,554 | 18,092 |
| Total | 354,399 | 351,883 | 2516 |

As shown in Table 2, there was a difference in the quantity of each soil layer. In the case of soil and weathered soil and weathered rock, we confirmed that the quantity measured based on GPS was underestimated compared with the actual quantity calculated from UAV photogrammetry. In the case of soft and hard rock, the quantity measured via GPS was overestimated. The impossibility to accurately predict the quantity of soft and hard rock located at the lowest point of the site in the design and procurement phase was shown. This is because the estimated volume was based on surface layers per soil type resulting from the interpolation of limited boring test results. This study provides new contextualization by comparing earthwork volume per soil type as well as comparing estimated volume with actual volume.

### 4.2. Cost Comparison of Computed Volume

Initially, BQ in the contract document used estimated volume based on GPS and the boring test result. By multiplying the volume by the unit price per soil type, the total estimated cost was USD 1,253,653. After the excavation, the actual volume was calculated using two sequential DSMs resulted from UAV photogrammetry. The actual cost stated in the final BQ was USD 1,070,595. There was a USD 183,057 (15.8%) cost difference, as shown in Table 3. A significant difference in the quantity of soft and hard rock between the two methods and the high unit price for vibration control blasting contributed to the cost difference. This finding is in accordance with Akgul et al. [5] who said UAV photogrammetry directly assists the success of earthwork estimation and construction costs. This study provides further analysis as it calculated earthwork cost per soil type. Accordingly, stakeholders can analyze the cost and develop countermeasure to avoid cost overrun.

**Table 3.** Cost comparison of the volume computed using GPS and UAVs.

| Soil Layer | Unit Price (USD/m$^3$) | Quantity (m$^3$) | | Cost (USD) | | Cost Difference (USD) | Remarks |
|---|---|---|---|---|---|---|---|
| | | **GPS** | **UAV** | **GPS** | **UAV** | | |
| Soil and weathered soil | 1.067 | 222,541 | 235,103 | 237,451 | 250,855 | −13,404 | |
| Weathered rock | 2.311 | 52,212 | 55,226 | 120,662 | 127,627 | −6965 | |
| Soft and hard rock | 11.244 | 79,646 | 61,554 | 895,540 | 692,113 | 203,426 | Vibration control blasting |
| Total | | 354,399 | 351,883 | 1,253,653 | 1,070,595 | 183,057 | |

Note: Exchange rate: 1125 Won/USD as of 16 February 2021.

### 4.3. Height Difference and Slope Monitoring

Following Jiang et al. [11], height difference and slope monitoring was conducted to monitor the progress and to analyze deformation of earthwork slope. During excavation, slopes deform and even become unstable, posing a risk. Slope monitoring can be used to assess slope stability and analyze its safety. For example, we chose a segment of interest for slope monitoring as shown in Figure 8. This segment was chosen because this excavation area required blasting equipment resulting in a high risk of slope deformation. On top of that, this location was located under high terrain and there was dense high residential area on top of the terrain. Results of this study show that from 24 September 2021 (green line) to 21 October 2021 (orange line), the bottom of the surface descends 3.427 m, which is consistent with the actual situation. Regarding slope displacement, there was a 12 cm displacement on average.

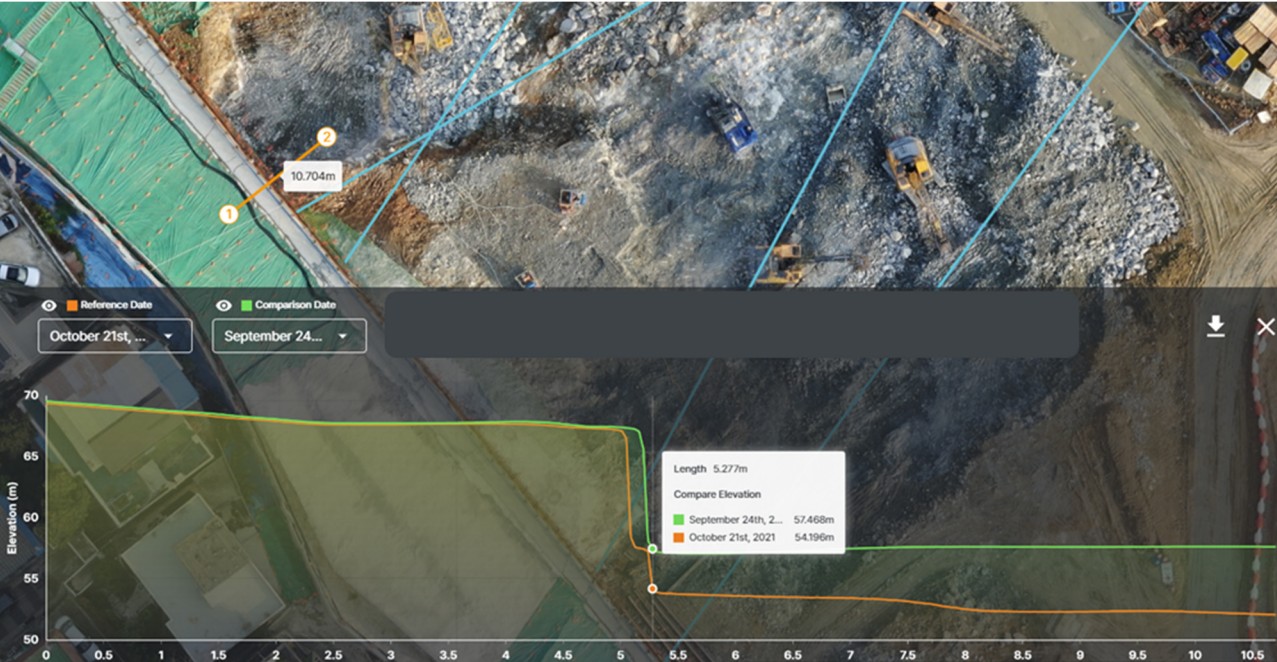

**Figure 8.** Height difference and slope monitoring of segment of interest.

### 4.4. Site Monitoring

Many construction projects combine diverse processes and activities, resulting in great dynamism and complexity. Stakeholders conduct meetings periodically to discuss work order, construction methods, equipment, and human resources arrangements to perform the construction work without defects or losses. Therefore, it is important to determine the overall situation of the construction site as the main reference in the meeting, which can be determined using UAV. UAV can provide high resolution and a real-time aerial view of the construction site, allowing stakeholders to monitor the site efficiently. For example, Figure 9a shows the UAV aerial photograph overlaid with foundation and ground anchor design. Stakeholders used this photograph to analyze the situation. The position of the foundation was moved to the side of the new excavation area as shown in Figure 9b. In this way, stakeholders can discuss, develop countermeasures, and revise drawings to proceed with construction dynamically.

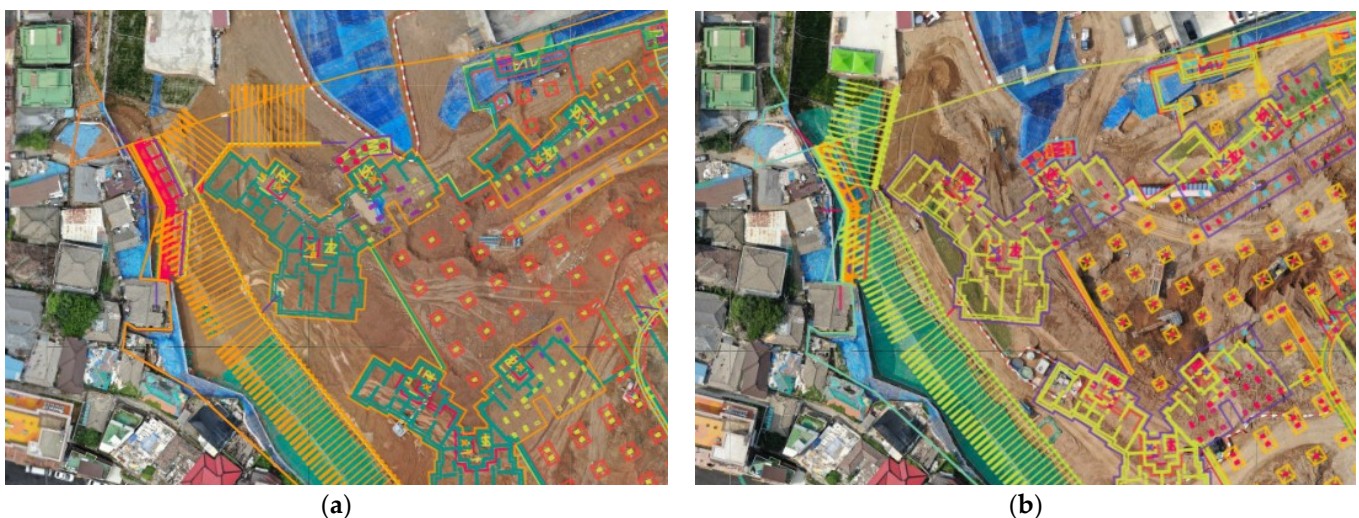

|  |  |
|---|---|
| (**a**) | (**b**) |

**Figure 9.** Site monitoring using UAV aerial photograph: (**a**) initial design of ground anchor; (**b**) changed design of ground anchor.

### 4.5. Documentation

Buildings are usually constructed over the course of years. On a daily basis, the changes are not significant. However, the changes become significant when we observe the construction progress over the span of months or years. It is important to archive this progress for several reasons: (1) to monitor the progress, (2) to serve as foundational data for the progress report, and (3) to prove the work that was performed in case there is a claim by other stakeholders. In this case study, the 3D point cloud and ortho-mosaic from the data processing stage were archived periodically (3–5 days). The 2D/3D models were then visualized in the UAV platform. One can compare the 2D/3D models from different times using the comparison function. A comparison of the construction site between two different times was thus visualized, as shown in Figure 10. This comparison was possible because of the site's UAV-based documentation.

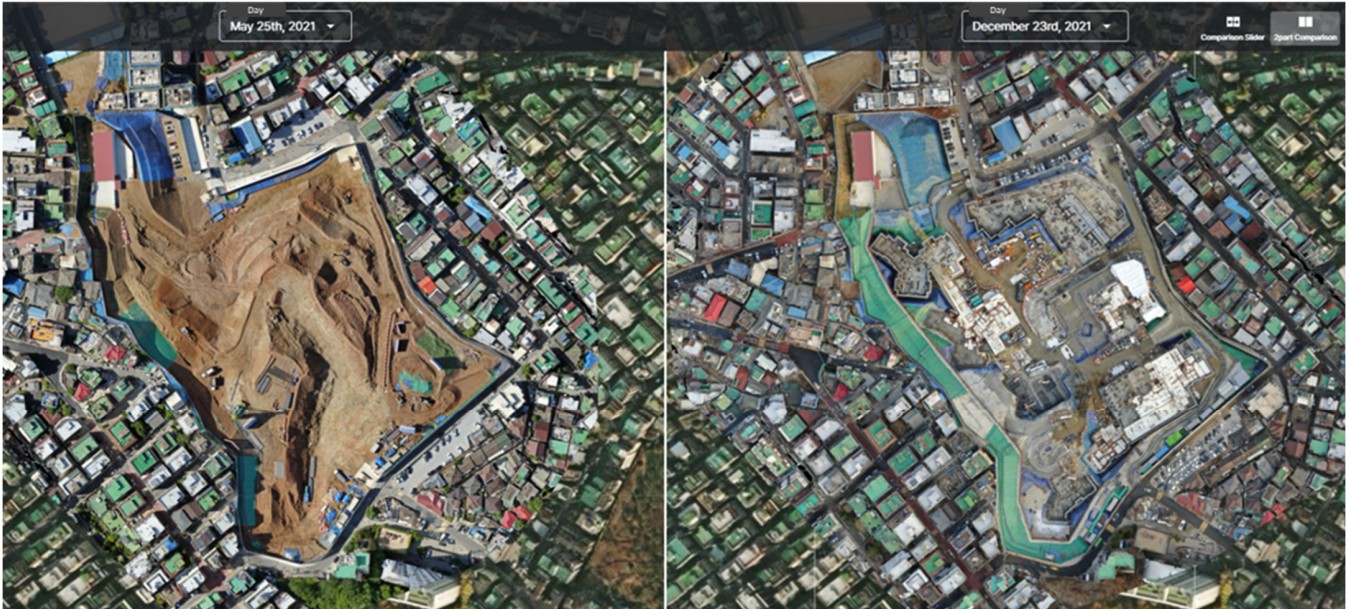

**Figure 10.** Documentation of earthwork activities at different times.

## 5. Conclusions

This study presents a framework for the UAV-based earthwork management of building projects. The entire framework includes data collection, data processing, and data analysis. This method produces a 3D terrain model, earthwork volume calculation, cross-section view, and 2D/3D visualization. It serves many functions, including progress monitoring by generating cut-and-fill volume and height difference reviews of two terrains from different times, aerial site monitoring, and documentation.

This framework was validated using an existing construction project in Seoul, Republic of Korea. In the case project, a volume comparison based on GPS and UAVs was conducted resulting in 2516 m$^3$ (0.71%) difference. We confirmed that the volume of soil and weathered soil and weathered rock, which have relatively low earthwork unit prices, increased, while the quantity of soft and hard rock, which is 5 to 11 times more expensive than the aforementioned types of rock, decreased, reducing the total construction cost by USD 183,057. These findings show UAV as an effective tool to monitor the actual excavated volume per soil type during construction stage. In this way, the potential conflict over earthwork quantity can be clearly resolved. As the information in the BQ is up-to-date, accurate, and transparent, UAV data can lead to fair deals between stakeholders.

The limitation of this study is that it only compared the final actual volume with the estimated volume. In next study, we will develop an earthwork cost and time simulation model based on UAV during the construction period to avoid risk of cost overrun and time delays of the project. Despite this limitation, UAV technology also contributed to continuous site monitoring to prevent risks that can occur during the earthwork process. By automating the earthwork digitization process, we believe that the proposed framework based on UAV will contribute to smart construction.

**Author Contributions:** Conceptualization, H.C.P., T.S.N.R. and S.K.; methodology, H.C.P., T.S.N.R. and S.K.; validation, H.C.P. and S.K.; formal analysis, H.C.P., T.S.N.R. and S.K.; investigation, H.C.P.; data resources, H.C.P.; writing—original draft preparation, T.S.N.R.; writing—review and editing, H.C.P., T.S.N.R. and S.K.; supervision, S.K.; project administration, S.K.; funding acquisition, H.C.P. and S.K. All authors have read and agreed to the published version of the manuscript.

**Funding:** This work was supported by the National Research Foundation of Korea (NRF) grants funded by the Korea government (MOE) (No. 2017R1D1A1B04033761 and No. 2022R1A2C2005276).

**Institutional Review Board Statement:** Not applicable.

**Informed Consent Statement:** Informed consent was obtained from all subjects involved in the study.

**Data Availability Statement:** Data sharing is not applicable to this article.

**Conflicts of Interest:** The authors declare no conflict of interest.

## Abbreviations

BQ      Bill of quantities
DSM     Digital surface model
GCPs    Ground control points
GPS     Global positioning system
UAV     Unmanned aerial vehicle
TS      Total station

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
