# Peer review of "UAV-Based High-Rise Buildings Earthwork Monitoring—A Case Study"

_sustainability, doi:10.3390/su141610179_

Round 1

Reviewer 1 Report

The paper is interesting. The paper is well structured, and the concept has been elaborated significantly.

I have a criticism that should be addressed regarding this paper. In the paper's title and throughout the text, the authors assert that they are utilising the UAV to manage the earthwork. While managing a construction project, it is necessary to take into account a variety of factors and select the most advantageous outcome using decision-making and optimisation techniques. In this study, UAVs are utilised for data collection, not activity management. After collecting data with an unmanned aerial vehicle, we use the gathered information to make a decision. Consequently, please address this comment throughout the entire paper and in the paper's title.

thanks.

Author Response

The title is changed from “Earthwork Management in a High-Density Residential Area using UAV technology - A Case Study” to “UAV-Based High-Rise Buildings Earthwork Monitoring - A Case Study”. The content of the paper has been adjusted according to the tittle.  

Reviewer 2 Report

The authors have conducted a case study on application of Unmanned Aerial Vehicle (UAV) in earthwork management in the high-density residential area. On the basis of UAV technique, the framework can be used to monitor the progress of cut-and-fill volume during a construction. The quantities obtained from the UAV approach are compared with those from the GPS approach. Indeed, the UAV approach is an excellent tool with many advantages, such as 3D terrain model, earthwork volume, cross-section view, and 2D/3D visualization, in the context of the earthwork management. However, this reviewer does not find enough novelty in the manuscript. It is admitted that the UAV approach is a promising tool. It is suggested to focus on technical details rather than on brief introductions of its functions.

Reviewer 3 Report

The paper describes the use of UAV photogrammetry for the volume calculation of an earthwork in Seoul. Despite the interesting work, I think that this study lack of originality, it does not add anything new to the existing methods and the results and discussion are not adequately presented.

Major comments:

At first, I found the state of art very repetitive (just as example, lines 45-49 with 150-153). I would summarize the concepts and emphasize what has been done in literature, advantages and disadvantages, what are you presenting as new at the end of the state of art.

In addition, I would not write a specific paragraph to explain the bill of quantities in earthwork management. It is a well-known practice, which can be summarized in the state of art.

Line 42-44: The UAV does not collect images. UAV is the means of transport on which a sensor is mounted. Moreover, the DTM is one of the results of photogrammetric technique. It would be better to write "the collected images are processed using the photogrammetry technique, which provides, within the results, a DTM of the area". Please, explain better these concepts, as the terminology is not very appropriate. Moreover, at the end you have also used the orthomosaic so it is better to add it here. The results you are interested in are both the DTM and the orthomosaic.

Line 46: I do not agree that the results are "more detailed" than TS or GNSS (GPS is not used alone anymore, so please check in all the paper). Maybe it can provide a global result compared to the discrete and isolated points obtained by TS and GNSS, but at the present days, the accuracy is generally lower than the traditional techniques. Please, clarify better the concept and what you would say with "more detailed results".

From line 169: are you sure is it only GPS and not GNSS? Please double check.

Lines 171-172: What do you mean by "boring test results"? Moreover the sentence is not properly written in English. Please specify better.

Lines 224-229: Were the GCP's taken by GNSS? Please specify also the modality. Moreover, have you used only GCP? How did you check the accuracy? It is a common practice to use many of the GCPs as Check Points (CPs) to check the accuracy of the results. If you did not, I think this is an important gap that must be filled. If you did, please provide a detailed table with the final accuracy both of the GCPs and the CPs.

Paragraph 4.1: does the website provide more information about how it performs the volume calculation? Or, at least, with which accuracy? I think this is important to specify, as it may have influence on the final results.

Paragraph 4.3: I agree that this is an important step, but what it is not clearto me is what is the advantage in using the UAV photogrammetry products for the site monitoring. What you have written can be also checked by a simple aerial photograph.

The results presented in the discussion are compared without any reference to the accuracy of the methods, so it is very difficult to understand which one of the two technologies is more efficient than the other. Moreover, there is no comparison with the state of art, what it is new in your work compared to what is already present in literature.

Minor comments:

Line 11: "to complete the earthwork on time" --> replace "earthwork" with "it".

Line 61: "comprises technical and management work". Delete the first "work".

Line 66-69: the English here is not very clear. Please rewrite the sentence.

Line 120: Party?? Do you mean "part"?

Line 166: the caption of Figure 2 should be "Case study of...."

Line 207: there is a bracket not closed.

Lines 218-219: the sentence is unnecessary.

Line 259: It would be better to specificy in bibliography the website.

Figure 8: in the second box, it would be better "Choose the area of interest.." The same in figure 10 "segment of interest" and in line 289.

Lines 304-305: replace the second "UAVs" with "it".

Line 321: what do you mean with "prove of work"?

Figure 14, second box: There is a typing error on "Archieve" which is "Archive"

Line 341: "the measured quantity"

Line 400: "The potential conflict"

Conclusions: Are there any future developments?

Paragraph 5.3: I think these general remarks can be added in the conclusions. Moreover, writing "The main contractor and subcontractor may sit together and check the quantity through the UAV platform. All parties trust the volume calculation based on UAV as the accuracy and transparency are high. In this way, the disagreement about the volume calculation is avoided." has a very poor scientific soundness. The concept can be explained is a more general way.

Author Response

Please see the attahed file.

Reviewer 4 Report

It is a very good paper with research outcomes that can have good impact on the industry. Few suggestions are added to improve the rigour of the article:

In section 1, line 45: authors are encouraged to replace word "cheaper" with something like "cost effective" or ""saving in cost". 

Line 50: Knowledge Gap requires supporting citation.

Add source of figure 1.

Section 2.2 of literature review needs more citations to support discussion.

Section 3.2.2 on data processing, needs more references to justify processes adopted like matching coordinates etc. to add vigour to the process adopted in the research.

Support your argument in line 303 and 304.

From section 5.1 to section 5.4, where possible try to support your findings with some previous research done in this area.

Mentions "Limitations of the Study"

Elaborate the impact of your study on construction industry, and the construction industry stake holders who will benefit form this research. It will be good to add recommendations for future research in this area.

Author Response

Please see the attahed file.

Round 2

Reviewer 2 Report

The authors have addressed my comments from the previous version.